# Expression of S100A16 Is Associated with Biological Aggressiveness and Poor Prognosis in Patients with Bladder Cancer Who Underwent Radical Cystectomy

**DOI:** 10.3390/ijms241914536

**Published:** 2023-09-26

**Authors:** Hiroki Katsumata, Kazumasa Matsumoto, Kengo Yanagita, Yuriko Shimizu, Shuhei Hirano, Kazuki Kitajima, Dai Koguchi, Masaomi Ikeda, Yuichi Sato, Masatsugu Iwamura

**Affiliations:** 1Department of Urology, Kitasato University School of Medicine, 1-15-1 Kitasato, Minami-ku, Sagamihara 252-0374, Kanagawa, Japan; giri_giri_zin_zin@yahoo.co.jp (H.K.); yulico@med.kitasato-u.ac.jp (Y.S.); s.hirano@med.kitasato-u.ac.jp (S.H.); kazurologist@gmail.com (K.K.); dai.k@med.kitasato-u.ac.jp (D.K.); ikeda.masaomi@grape.plala.or.jp (M.I.); sato.yuichi@kobal.co.jp (Y.S.); miwamura@med.kitasato-u.ac.jp (M.I.); 2Biofluid Biomarker Center, Niigata University, 8050 ikarashi 2-no-cho, Nishi-ku, Niigata 950-2181, Niigata, Japan; yanagita-bbc@ccr.niigata-u.ac.jp; 3KITASATO-OTSUKA Biomedical Assay Laboratories Co., Ltd., 1-15-1 Kitasato, Minami-ku, Sagamihara 252-0329, Kanagawa, Japan

**Keywords:** bladder cancer, cystectomy, S100A16 protein, AHNAK nucleoprotein 2, urothelial carcinoma, immunohistochemistry

## Abstract

S100 calcium binding protein A16 (S100A16) is expressed in various cancers; however, there are few reports on S100A16 in bladder cancer (BC). We retrospectively investigated clinical data including clinicopathological features in 121 patients with BC who underwent radical cystectomy (RC). Immunohistochemical staining was performed to evaluate S100A16 expression in archived specimens. Cases with >5% expression and more than moderate staining intensity on cancer cells were considered positive. S100A16 expression was observed in 54 patients (44.6%). Univariate analysis showed that S100A16 expression was significantly associated with age, pT stage, recurrence, and cancer-specific death. Kaplan–Meier analyses showed that patients with S100A16 expression had shorter overall survival (OS), cancer-specific survival (CSS), and recurrence-free survival (RFS) than those without S100A16 expression. In multivariate analysis, pT stage was an independent prognostic factor for OS and lymph node metastasis for CSS and RFS. S100A16 expression may be a biomarker of a biologically aggressive phenotype and poor prognosis in patients with BC who underwent RC. The PI3k/Akt signaling pathway is probably associated with S100A16 and may be a therapeutic target.

## 1. Introduction

Bladder cancer (BC) is the 10th most common cancer worldwide, with approximately 570,000 new cases and 210,000 deaths annually. The morbidity and mortality rates of BC are 3.0 and 2.1 per 100,000, respectively [1]. Upon the initial diagnosis, 75% of BC cases are non-muscle-invasive bladder cancer (NMIBC) and treated with transurethral resection of the bladder tumor (TURBT) [2]. Thereafter, according to the risk classification, intravesical chemotherapy treatment with Bacillus Calmette-Guerin (BCG) or a second TURBT is performed. However, in approximately 20% of patients, the disease progresses to MIBC [3]. Although radical cystectomy (RC) is the gold-standard treatment for MIBC, the 5-year overall survival (OS) rate is modest due to postoperative local recurrence and distant metastasis [4,5,6].

Prognostic risk factors related to surgery have been investigated to improve such unfavorable outcomes following RC. The tumor grade, lymphovascular invasion (LVI), and lymph node status are prognostic risk factors [7]. These prognostic variables are helpful in estimating the recurrence risk and survival outcomes; however, they cannot sufficiently help predict the individual prognosis and determine an appropriate treatment for individual patients. While molecular biomarkers are used to estimate the efficacy of drug therapy and improve the survival of patients with various other cancers such as breast cancer [8], the identification of a novel biomarker for BC is urgently needed. Many biomarkers such as urinary microRNA have been evaluated as potential biomarkers of BC, but none of them have been established to replace cystoscopy and cytology. The study of biomarkers is a developing landscape [9,10,11].

S100 proteins are the largest family of calcium-binding proteins of the EF–hand type, consisting of more than 25 members, which undergo conformational changes upon calcium binding and exhibit various intracellular regulatory activities [12,13,14,15]. S100 calcium binding protein A16 (S100A16), which is the most recent member of the S100 family, promotes adipogenesis and is involved in the suppression of calcium-induced weight gain [16,17].

S100A16 is associated with several cancer types, particularly tumor progression and poor prognosis in lung, prostate, and breast cancers [18,19,20]. In prostate cancer, S100A16 promotes cell proliferation and metastasis via protein kinase B (Akt) and the extracellular signal-regulated kinase signaling pathway [19]. In breast cancer, the overexpression of S100A16 promotes cell proliferation, colony formation, tumor cell invasion, and migration through the epithelial–mesenchymal transition (EMT) pathway and transcription factors such as Notch1 and zinc finger E-box-binding homeobox 1/2 [20]. The downregulation of thiosulfate transferase in BC promoted tumor invasion, migration, and the EMT, whereas the downregulation of S100A16 suppresses these processes [21]. Elucidating the mechanism of action of S100A16 in BC will confirm its role as a tumor-promoting factor. S100A16 is expected to serve as a biomarker for the aggressive tumor phenotype.

The purpose of this study was to investigate the expression of S100A16 in BC tumor cells by immunohistochemistry and to evaluate the tumor aggressiveness and its prognostic impact on patients with BC who underwent RC.

## 2. Results

### 2.1. Immunohistochemistry

Figure 1 shows S100A16 staining in the tumor tissue of study groups. Most S100A16 staining occurred in the plasma membrane. S100A16 was not detected in normal urothelial cells. In total, 54 of 121 patients (44.6%) showed positive S100A16 staining.

### 2.2. Association of S100A16 Expression with Clinicopathological Characteristics

Table 1 summarizes the clinicopathological characteristics. A total of 58 patients were still alive at the end of the follow-up, 47 patients died of BC, and 16 patients died from other causes. Age at cystectomy, pT stage, recurrence, and cancer-specific death were associated with S100A16 expression. The other factors did not lead to significant differences in S100A16 expression.

The association of AHNAK nucleoprotein 2 (AHNAK2) expression with S100A16 is shown in Table 2. The expression of S100A16 was associated with the positive status of AHNAK2 (*p* = 0.008).

### 2.3. Survival Outcomes

Kaplan–Meier analysis showed that patients with S100A16 expression had shorter OS, CSS, and RFS than those without expression (*p* = 0.003, *p* = 0.0042, and *p* = 0.0035, respectively; Figure 2). Univariate Cox regression analysis showed that S100A16 expression, lymph node status, LVI, and pT stage were significant predictors of patients’ prognosis, including OS, CSS, and RFS (Table 3). In multivariate Cox regression analysis, only pT stage was an independent prognostic predictor in OS. Lymph node metastasis was the only independent prognostic predictor of CSS and RFS.

## 3. Discussion

RC is the gold standard treatment for patients with MIBC and BCG refractory to NMIBC. About 50% of patients experience disease recurrence, and this high recurrence rate is a critical issue [7]; however, clinical and pathological variables cannot sufficiently predict individual prognosis. Thus, there is an urgent need for novel biomarkers that can play a predictive role in determining optimal treatment strategies for individuals.

In this study, we retrospectively investigated the prognostic impact of S100A16 expression on patients with BC treated with RC. S100A16 expression was significantly correlated with age, pT stage, recurrence, and cancer-specific death. Kaplan–Meier analysis showed that S100A16 expression increased the risk of probability of CSS and RFS. These results indicate that S100A16 may be a prognostic biomarker for BC.

There are some studies showing that S100A16 induces EMT in several cancer types [22]. EMT is a process in which epithelial cells lose polarity and cell–cell adhesions and progress to invasive mesenchymal cells. EMT is associated with the metastatic potential of several types of epithelial cancer and is thought to play an important role as a tumor promoter [23]. Zhou et al. [20] reported that S100A16 promotes EMT in breast cancer. Li et al. [24] showed that S100A16 induces EMT to promote metastasis in pancreatic cancer. Currently, evaluations of S100A16 and EMT in BC have been limited. Analyses based on BC cell lines have shown that Snail, the EMT-related transcription factor, regulates S100A16 [25]. The knockdown of S100A16 was shown to reduce the expression of N-cadherin, vimentin, and slug and increase the expression of E-cadherin in BC cells [10,26]. S100A16 may directly or indirectly act to EMT associated marker and induce EMT. EMT correlated with more aggressiveness and chemoresistance among malignant tumors and induced poor prognosis. The shorter period of CSS and RFS in this study could be attributed to tumor aggressiveness by EMT; however, the association of S100A16 and EMT in BC has not been elucidated. Further research may be warranted.

The present study showed a significant correlation between S100A16 and AHNAK2. The association of S100A16 and the phosphoinositide 3-kinase (PI3K)/Akt signaling pathway has been reported. The PI3K/Akt signaling pathway is one of the most important intracellular pathways, which regulates cell survival, cellular growth, and cell cycle progression. This pathway induces the progression of tumor cells and plays a key role in tumor proliferation, invasion, and chemotherapy resistance [27]. Zhang et al. [28] reported that S100A16 can promote tumor proliferation, migration, and angiogenesis by regulating the PI3K/Akt signaling pathway in cervical cancer. Li et al. [29] showed that S100A16 suppresses apoptosis and promotes cell proliferation via the Akt signaling pathway in pancreatic cancer. These studies suggest that S100A16 is a possible tumor promoter via the PI3K/Akt signaling pathway. In our previous report, AHNAK2 was significantly increased in BC specimens compared to normal bladder tissue [30]. Li et al. [31] showed that AHNAK2 promotes cell proliferation, migration, and invasive abilities via the PI3K/Akt signaling pathway in uveal melanoma. Taken together, these findings show that PI3K/Akt signaling is a common tumor-promoting pathway for S100A16 and AHNAK2. According to their molecular expression, the PI3K/Akt pathway may play a role in the tumor aggressiveness of BC. 

Some cancer therapies based on the PI3K signaling pathway are undergoing clinical investigation [32]. Qu et al. [33] reported that therapies targeting the PI3K/Akt signaling pathway in hepatocellular carcinoma showed anticancer effects by promoting autophagy and the apoptosis of tumor cells. If a correlation between S100A16 and the PI3K/Akt signaling pathway is established, targeting S100A16 may be useful for personalized therapy in patients with BC.

An association between S100A16 and chemotherapy resistance was shown in an in vitro study. Wang et al. [25] showed that S100A16 is upregulated in mitomycin C-resistant BC cell lines compared with normal BC cell lines. In lung adenocarcinoma, S100A16 was found to be a prognostic marker for platinum-based adjuvant chemotherapy in an immunohistochemical study [34]. However, our study did not show a significant correlation between S100A16 and response to chemotherapy. Recently, immune checkpoint inhibitors (ICIs) and enfortumab vedotin (EV) have been made available to patients with BC. Nivolumab is used for adjuvant chemotherapy (AC), and pembrolizumab and EV are used as salvage chemotherapy (SC) [35]. However, our present cohort included only a few cases treated with new treatment modalities. Future studies will collect these cases and investigate the association of S100A16 expression and the response to ICI or EV in patients with BC treated with RC.

This study had several limitations. First, it was a single-center retrospective analysis with selection bias. Second, the sample size was small. Notably, there were few cases in which chemotherapy was administered, and it is possible that the results would differ depending on the cases analyzed. Third, RC was performed by several different surgeons, and additional therapy such as AC and SC was decided by each doctor; these differences may have influenced the results. Finally, chemotherapy did not include ICI or EV; thus, future analyses should include these new treatments.

## 4. Material and Methods

### 4.1. Patients

We retrospectively analyzed the clinical data and archived specimens from 167 patients with BC who underwent RC between 1990 and 2017 at Kitasato University Hospital (Kanagawa, Japan). We excluded 10 patients who had histological variants of BC including squamous cell carcinoma, adenocarcinoma, and small cell carcinoma; 22 who had been previously treated with neoadjuvant chemotherapy; and 14 who were lost to follow-up. We obtained normal urothelial specimens from adjacent normal bladder tissue with NMIBC in the cohort. The study group comprised 97 men (80%) and 24 women (20%), with a median age of 65 years (range: 40–82 years). No patients received preoperative neoadjuvant chemotherapy or radiotherapy, and no distant metastasis was observed at the time of diagnosis. RC was performed in patients with pathologically proven MIBC and in those with NMIBC who failed to respond intravesical therapy [36]. Patients’ characteristics were obtained from medical records including age at the time of surgery, sex, tumor grade, concomitant carcinoma in situ (CIS), pathological status including pT stage and pN stage, LVI status, and history of AC and SC. AC was performed for patients with ≥pT3 or for those with a positive lymph node status [37]. The response of AC or SC was evaluated by the Response Evaluation Criteria in Solid Tumors (RECIST) version 1.1. We categorized the patients as either responsive (complete response or partial response) or nonresponsive (stable disease or progression disease). Pathological staging was evaluated according to the 2002 TNM classification for assessment. Pathological grading was assessed according to the 1973 World Health Organization classification. The median follow-up time was 33.7 months (mean 70.0 months, range: 0.7–288.3 months). This retrospective study confirmed the Recommendations for Tumor Marker Prognostic Studies (REMARK) guideline [38] (Appendix A). The ethics committee of Kitasato University School of Medicine approved the study, and opt-out was obtained from the patients (B17-010). The patients could refuse study entry and discontinue participation at any time.

### 4.2. Immunohistochemistry and Scoring

Three-micron-thick sections from 10% formalin-fixed and paraffin-embedded BC cell lines and 121 surgically resected BC cell were deparaffinized in xylene, rehydrated in a descending ethanol series, and then treated with 3% hydrogen peroxide for 10 min. After the antigen was retrieved by autoclaving in 0.01 mol/L citrate buffer (pH 6.0) with 0.1% Tween 20 at 121 °C for 10 min, the sections were reacted with 1000-fold-diluted anti-S100A16 polyclonal antibody (ab130419; Abcam PLC, Cambridge, UK) for 16–18 h at 4 °C. After rinsing in Tris-buffered saline three times for 5 min each, the sections were reacted with ChemMate Envision reagent (Dako) for 30 min at room temperature. Finally, the sections were visualized with Stable DAB solution (Invitrogen, Carlsbad, CA, USA) and counterstained with Mayer’s hematoxylin. 

Immunohistochemistry evaluated both the staining intensity and percentage of positive tumor cells. The staining intensity was categorized into four groups: 0 = negative; 1 = weak; 2 = moderate; and 3 = strong. The tumor with a staining score of 2 or 3 and tissue consisting of >5% S100A16 expression in the tumor cell membrane was considered positive [34]. Only normal urothelial tissues were set as a control group for immunohistochemistry. All immunostained sections were reviewed by two investigators (H.K. and Y.S.) without any knowledge of the clinical data. Discordant cases were reviewed and discussed until a consensus was reached. 

AHNAK2 (HPA004145; Sigma-Aldrich, St. Louis, MO, USA) immunohistochemical staining was performed as previously described [39]. To better understand the mechanism of action of S100A16 in BC, we analyzed the association between S100A16 and AHNAK2. The current study cohort and the previous AHNAK2 cohort generally overlapped, and 110 of the cases evaluated in the AHNAK2 study were used to assess correlations. AHNAK2 expression scores were categorized as scores < 3 (low expression) or ≥ 3 (high expression) using the sum index score.

### 4.3. Statistical Analysis

In the immunohistochemistry analysis, age (<65 vs. ≥65), pathological stage (≤pT2 vs. ≥pT3), pathological grade (1 and 2 vs. 3), and lymph node status (N0 vs. N1 and N2) were evaluated as dichotomized variables. The association between S100A16 and clinicopathological status (sex, age, pathological stage, CIS, lymph node status, pathological grade, LVI, adjuvant chemotherapy, salvage chemotherapy, recurrence, cancer-specific survival, and AHNAK2) was evaluated by the Fisher’s exact test. OS, CSS, and RFS were estimated by the Kaplan–Meier method using log-rank tests. Univariate and multivariate analyses were performed using Cox proportional hazard analysis to estimate the association between S100A16 expression and clinicopathological variables. *p* < 0.05 was considered statistically significant. All reported p values are two-sided. Stata 17 for Windows (Stata, Chicago, IL, USA) was used for all analyses.

## 5. Conclusions

S100A16 expression is associated with a poor prognosis in patients with BC who underwent RC and is a possible biological marker of an aggressive phenotype. Based on the mechanism of action of AHNAK2, it is considered that S100A16 may be associated with the PI3K/Akt signaling pathway in BC. Therapies that target PI3K/Akt have been reported. The present study showed that S100A16 may have potential usefulness as a new biomarker for BC; further research is needed to confirm the role of S100A16 in BC.

## Figures and Tables

**Figure 1 ijms-24-14536-f001:**
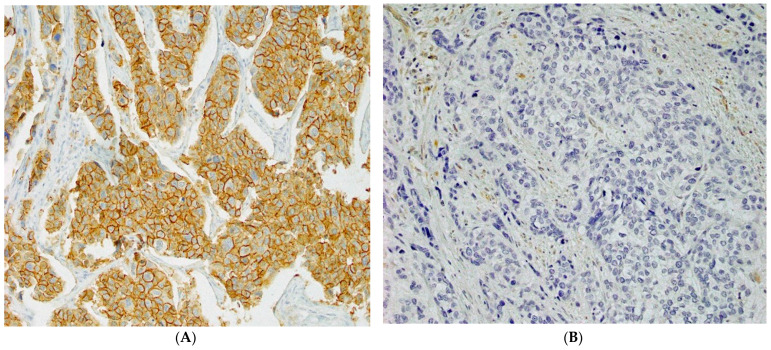
Immunohistochemical staining of S100A16 in bladder cancer. (**A**) Positive expression (score 3 × 4). (**B**) Negative expression (score 0 × 0). All 400× original magnification.

**Figure 2 ijms-24-14536-f002:**
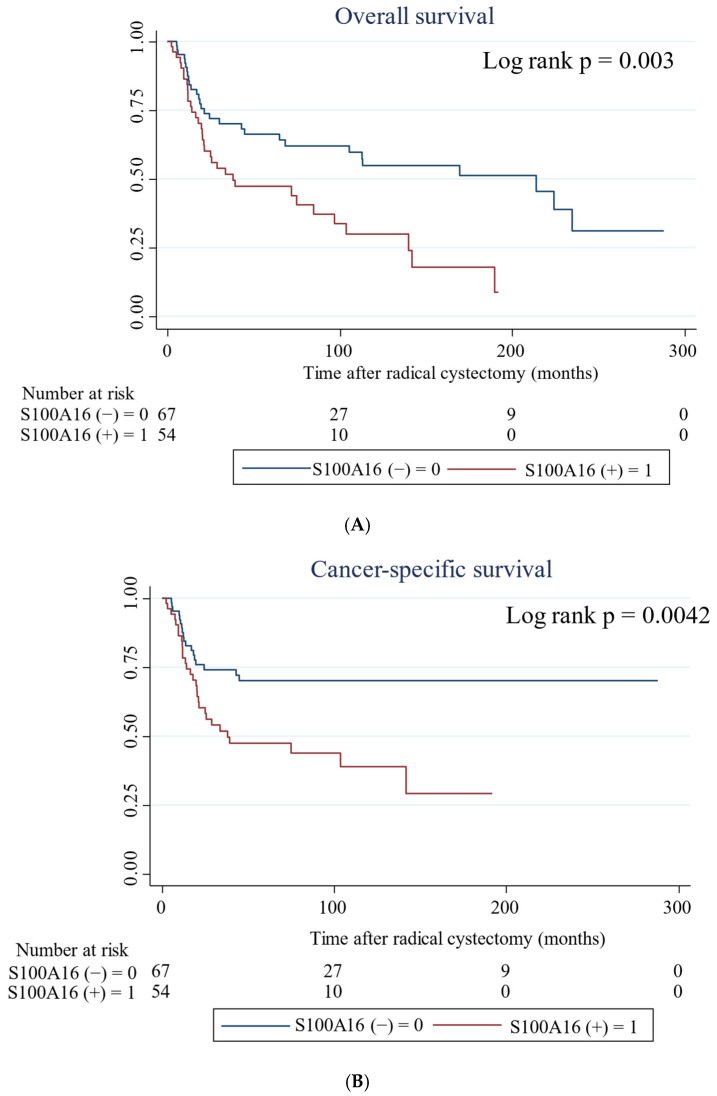
Probability of survival in patients with bladder cancer according to S100A16 expression estimated by Kaplan–Meier analysis. (**A**) Overall survival; (**B**) Cancer-specific survival; (**C**) Recurrence-free survival.

**Table 1 ijms-24-14536-t001:** The relationships between S100A16 expression and clinicopathological characteristics.

Characteristics		S100A16		
	Total No. (%)	Negative (%)	Positive (%)	*p*-Value
Overall	121	67 (55.4)	54 (44.6)	
Age, years				
Median (IQR)	65 (40–82)	63 (41–79)	67.5 (40–82)	
≤65	64 (52.9)	29 (43.3)	35 (64.8)	0.028
>65	57 (47.1)	38 (56.7)	19 (35.2)	
Sex				
Male	97 (80.2)	54 (80.6)	43 (79.6)	1
Female	24 (19.8)	13 (19.4)	11 (20.4)	
pTstage				
≤pT2	55 (45.5)	40 (59.7)	15 (27.8)	0.001
≥pT3	66 (54.5)	27 (40.3)	39 (72.2)	
Carcinoma in situ				
Negative	102 (84.3)	54 (80.6)	48 (88.9)	0.315
Positive	19 (15.7)	13 (19.4)	6 (11.1)	
Lymph node status				
N0	89 (78.1)	52 (82.5)	37 (72.5)	0.256
N+	25 (21.9)	11 (17.5)	14 (27.5)	
Pathological grade				
G1–2	52 (43.3)	28 (41.8)	24 (45.3)	0.715
G3	68 (56.7)	39 (58.2)	29 (54.7)	
LVI				
Present	67 (60.9)	34 (56.7)	33 (66.0)	0.334
Absent	43 (39.1)	26 (43.3)	17 (34.0)	
Adjuvant chemotherapy				
No recurrence	10 (38.5)	8 (53.3)	2 (18.2)	0.109
Recurrence	16 (61.5)	7 (46.7)	9 (81.8)	
Salvage chemotherapy				
Nonresponse	22 (73.3)	14 (73.7)	8 (72.7)	1
Response	8 (26.7)	5 (26.3)	3 (27.3)	
Recurrence				
No	63 (52.1)	41 (61.2)	22 (40.7)	0.029
Yes	58 (47.9)	26 (38.8)	32 (59.3)	
Cancer-specific death				
No	74 (61.2)	49 (73.1)	25 (46.3)	0.003
Yes	47 (38.8)	18 (26.9)	29 (53.7)	

No.: number; LVI: lymphovascular invasion; IQR: interquartile range.

**Table 2 ijms-24-14536-t002:** The association of AHNAK2 expression with S100A16.

	AHNAK2 Expression		
	Negative	Positive	*p*-Value
Total (%)	53 (48.2)	57 (51.8)	
S100A16			
Negative	35 (66.0)	23 (40.4)	0.008
Positive	18 (34.0)	34 (59.6)	

**Table 3 ijms-24-14536-t003:** Univariate and multivariate Cox proportional hazard analyses for predicting overall survival, cancer-specific survival, and recurrence-free survival in patients with bladder cancer treated with radical cystectomy.

**Overall Survival**						
	**Univariate**			**Multivariate**		
	**HR**	**95% CI**	***p*-Value**	**HR**	**95% CI**	***p*-Value**
S100A16	2.16	1.28–3.63	0.004	1.54	0.84–2.79	0.16
pN+	2.68	1.50–4.80	0.001	1.64	0.85–3.16	0.137
Grade3	1.85	1.08–3.14	0.024	1.39	0.77–2.52	0.28
LVI present	2.26	1.26–4.05	0.006	1.15	0.58–2.28	0.679
CIS	0.3	0.11–0.83	0.02	0.43	0.14–1.26	0.123
pTstage	5.08	2.72–9.46	0	3.65	1.80–7.39	0
**Cancer-specific survival**						
	**Univariate**			**Multivariate**		
	**HR**	**95% CI**	***p*-Value**	**HR**	**95% CI**	***p*-Value**
S100A16	2.31	1.28–4.18	0.005	1.71	0.84–3.46	0.136
pN+	3.16	1.69–5.91	0	2.19	1.07–4.48	0.033
Grade3	1.64	0.89–3.01	0.111	1.31	0.66–2.62	0.445
LVI present	2.53	1.24–5.16	0.01	1.34	0.58–3.10	0.494
CIS	0.32	0.10–1.04	0.059	0.47	0.14–1.59	0.225
pTstage	3.41	1.76–6.59	0	2.02	0.94–4.36	0.07
**Recurrence-free survival**						
	**Univariate**			**Multivariate**		
	**HR**	**95% CI**	***p*-Value**	**HR**	**95% CI**	***p*-Value**
S100A16	1.74	1.03–2.92	0.037	1.44	0.77–2.70	0.258
pN+	3.03	1.72–5.35	0	1.97	1.02–3.82	0.044
Grade3	1.46	0.85–2.51	0.165	1.23	0.65–2.32	0.529
LVI present	2.41	1.28–4.53	0.006	1.66	0.77–3.57	0.193
CIS	0.55	0.23–1.27	0.162	0.89	0.36–2.20	0.807
pTstage	2.88	1.63–5.08	0	1.78	0.90–3.53	0.097

CI: confidence interval; HR: hazard ratio; LVI: lymphovascular invasion; CIS: carcinoma in situ.

## Data Availability

The datasets used and/or analyzed during the study are available from the corresponding author on reasonable request.

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
