# Peer review of "Expression of S100A16 Is Associated with Biological Aggressiveness and Poor Prognosis in Patients with Bladder Cancer Who Underwent Radical Cystectomy"

_ijms, 2023, doi:10.3390/ijms241914536_

Round 1

Reviewer 1 Report

Very interesting topic and results, however, several issues require clarification: 1) who cwas included to the control group 2) Figure 1 presents the results of positive and negative expression but there is no information about the sample, whether from the control or study group 3) prognostic markers should be reported in accordance with the Reporting Recommendations for Tumor Marker Prognostic Studies (REMARK) guidelines, and therefore why these guidelines were not followed.

Reviewer 2 Report

This study aims to investigate the expression of S100A16 in BC tumor cells by immunohistochemistry and to evaluate the tumor aggressiveness and its prognostic impact on patients with BC who underwent RC.

I believe that the study has sufficient merit to be considered for publication on International Journal of Molecular Sciences, although major revisions are required.

MAJOR COMMENTS

-       Abstract. The abstract provides a succinct summary of the study's objectives, methods, and key findings.

-       Introduction: Consider citing recent statistics on BC incidence and mortality for up-to-date context. In line 54 I suggest highlight the role of new biomarkers in bladder cancer. I recommend to the authors this reference that I think is important and that can be of great help when modifying the manuscript. (doi: 10.3390/ijms241310846, PMID 37446024).

-       Check typos in the text, many were found while reading.

-       Discussion: The correlation between S100A16 and chemotherapy resistance should be discussed in more detail. For example, what are the clinical implications of this discovery? Could it influence treatment decisions?

A more detailed explanation of S100A16's role in the context of bladder cancer and the EMT (epithelial-mesenchymal transition) is needed. Authors should discuss how S100A16 might influence this process and how it could be linked to prognosis.

-       Conclusion: It might be helpful to emphasize this study's specific contribution to understanding bladder cancer and the potential significance of S100A16 as a biomarker.

Moderate editing of English language required

Round 2

Reviewer 1 Report

The authors have not answered these points:

1) who cwas included to the control group 2) Figure 1 presents the results of positive and negative expression but there is no information about the sample, whether from the control or study group 

They added information that they included normal urothelium in the control group, but no information from whom and whether the bioethics committee consented to obtaining normal tissue.

Reviewer 2 Report

I believe that the study has sufficient merit to be considered for publication
